# PrAViC: Probabilistic Adaptation Framework for Real-Time Video Classification

## Abstract

Video processing is generally divided into two main categories: processing of the entire video, which typically yields optimal classification outcomes, and real-time processing, where the objective is to make a decision as promptly as possible. The latter is often driven by the need to identify rapidly potential critical or dangerous situations. These could include machine failure, traffic accidents, heart problems, or dangerous behavior. Although the models dedicated to the processing of entire videos are typically well-defined and clearly presented in the literature, this is not the case for online processing, where a plethora of hand-devised methods exist. To address this, we present PrAViC, a novel, unified, and theoretically-based adaptation framework for dealing with the online classification problem for video data. The initial phase of our study is to establish a robust mathematical foundation for the theory of classification of sequential data, with the potential to make a decision at an early stage. This allows us to construct a natural function that encourages the model to return an outcome much faster. The subsequent phase is to present a straightforward and readily implementable method for adapting offline models to the online setting with recurrent operations. Finally, PrAViC is evaluated through comparison with existing state-of-the-art offline and online models and datasets, enabling the network to significantly reduce the time required to reach classification decisions while maintaining, or even enhancing, accuracy.

## 1 Introduction

In recent years, there has been a notable increase in the utilization of convolutional neural networks (CNNs) across a range of fields where the capacity to make expeditious decisions could be crucial. This includes such fields as medicine (Krenzer et al., 2023; Sapitri et al., 2023), human action recognition (HAR) (Mollahosseini et al., 2016; Yang & Dai, 2023), and autonomous driving (Wu et al., 2017). However, despite the growing prevalence of CNNs in these domains, there remains a lack of a unified approach to the problem of making early decisions based solely on the initial frames.

Conversely, numerous offline approaches have been proposed (see, e.g., (Bhola & Vishwakarma, 2024; Karim et al., 2024; Kaseris et al., 2024; Ming et al., 2024; Yao et al., 2019)) to address the problem of video data classification. However, these models usually require access to entire videos, which precludes their real-time applicability. Although certain techniques have been developed to facilitate the adaptation of offline models to the online domain (e.g., those proposed by Köpüklü et al. (2022) or Xiao et al. (2023)), there remains a need for the development of more generalizable solutions that can accommodate diverse forms of data.

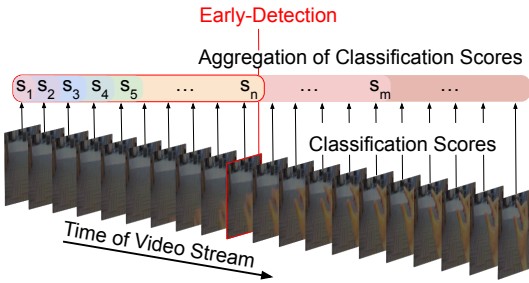

Figure 1: Our approach consists of three components: an online classifier generating Classification Scores, an Aggregation module, and an Early-Detection mechanism.

To address the aforementioned gap, we propose a novel probabilistic adaptation framework for real-time video classification (PrAViC), see Fig. 1. In contrast to traditional methodologies, our approach allows for the adoption of existing 3D CNN models, wherein subtle adjustments are made to leverage

the strengths of traditional CNN-3D networks while addressing specific challenges related to depth processing and feature extraction. Moreover, our strategy paves the way for recursive utilization. The implications of this technological development are far-reaching, impacting a multitude of domains including industry, medicine, and public safety. In these fields, the capacity to conduct rapid real-time analysis is of paramount importance for informed decision-making and the implementation of proactive measures.

In the experimental study, we demonstrate the efficacy of our approach when applied to a selection of state-of-the-art offline and online models trained on three real-world datasets, including two publicly available datasets, UCF101 (Soomro et al., 2012), EgoGesture (Zhang et al., 2018), Jester (Materzynska et al., 2019), and Kinetics-400 (Kay et al., 2017), as well as a closed-access real Ultrasound dataset, comprising Doppler ultra-317 sound images representing short-axis and suprasternal views of newborn hearts. Moreover, we introduce an innovative function that enables the model to make earlier exits (decisions) when sufficient evidence is accumulated. The impact of such a solution is illustrated in Fig. 2. By integrating this function into the training objective (see Eq. (5)), the model optimizes the timing of its decisions, enhancing efficiency with occasional slight loss of accuracy.

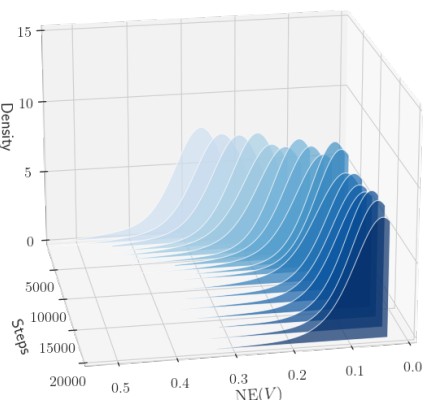

Figure 2: Progress of the model's early decision function during training.

In conclusion, our contributions can be summarized as follows:

- we introduce PrAViC, a novel, unified, and theoretically-based probabilistic adaptation framework for online classification of video data, which encourages the network to make a decision at an early stage,
- we propose a straightforward and easy to implement method for adapting offline video classification models to online use,
- we conduct experiments which indicate that the PrAViC is capable of facilitating earlier classification decisions while maintaining, or even improving, accuracy when compared to selected offline and online state-of-the-art solutions.

## 2 RELATED WORKS

While video-based 3D networks have been widely studied in offline settings (Bhola & Vishwakarma, 2024; Karim et al., 2024; Kaseris et al., 2024; Ming et al., 2024; Yao et al., 2019), developing online models remains still a challenge. Such online models are able to real-time classification, enabling applications such as emergency situations detection or medical diagnostics.

In consideration of the video as a series of consecutive frames, each frame may be classified on an individual basis through the application of 2D CNN models (Kumar, 2019). This approach represents an online technique and demonstrates effectiveness in real-time classification, due to the limited number of parameters involved. However, the absence of temporal information from the video as a whole may present a limitation, potentially leading to misclassifications such as predicting "sitting down" when the action involves transitioning from sitting to standing. Consequently, in order to develop a model capable of processing online data in real-time, researchers frequently elect to utilize 2D networks, complemented by supplementary mechanisms for the management of temporal data (Chang & Huang, 2024; Shen et al., 2023; Wang et al., 2021; Xiao et al., 2023; Xu et al., 2023). For instance, the authors of (Xiao et al., 2023) employed a temporal shift module (TSM). Their approach involves shifting a portion of the channels along the temporal dimension, thereby facilitating the capture of temporal relationships.

On the other hand, it has been demonstrated that 3D CNNs yield superior accuracy in video classification tasks compared to 2D CNNs (Carreira & Zisserman, 2017). Consequently, novel approaches have been devised that utilize distinct versions of 3D convolutional kernels. A number of studies have proposed the development of dedicated architectures for online 3D networks, including those

presented by Kim et al. (2024), Krenzer et al. (2023), Sapitri et al. (2023), and Yang & Dai (2023), which are designed to operate in real-time.

Another commonly utilized approach is the combination of 3D CNN architectures with various long short-term memory (LSTM) models. For example, Lu et al. (2024) introduce a DNN that merges 3D DenseNet variants and BiLSTM. In turn, Chen et al. (2023) propose a combination of R2plus1D and ConvLSTM in a parallel module. The proposed network utilizes the attention mechanism to extract the features that require attention in the channel and the spatial axes.

An alternative approach to the problem of online video classification, as proposed by Üstek et al. (2023), involves combining a vision transformer for human pose estimation with a CNN-BiLSTM network for spatio-temporal modelling within keypoint sequences. Similarly, attention mechanisms, in conjunction with transformer layers, have also been employed in (Huang et al., 2023).

The concept of converting various well-known resource-efficient 2D CNNs into 3D CNNs, as proposed by Köpüklü et al. (2020), is the most closely related to our approach. In this context, Köpüklü et al. (2022) investigate the potential of adapting 3D CNNs (particularly the 3D ResNet family of models) for online video stream processing. Their approach involves the elimination of temporal downsampling and the utilization of a cache to store intermediate volumes of the architecture, which can then be accessed during inference. A comparable solution has also been proposed by Hedegaard & Iosifidis (2022), who advanced the concept of weight-compatible reformulation of 3D CNNs, designated as Continual 3D Convolutional Neural Networks (Co3D CNNs). Co3D CNNs facilitate the processing of videos in a frame-by-frame manner, utilizing existing 3D CNN weights, thereby obviating the necessity for further finetuning.

# 3 PROBABILISTIC ADAPTATION FRAMEWORK FOR REAL-TIME VIDEO CLASSIFICATION (PRAVIC)

This section presents the details of the proposed PrAViC model (**Pr**obabilistic **A**daptation Framework for Real-time **Vi**deo **C**lassification). For the reader's convenience, the problem of video classification is divided into offline and online. In the offline case, where the entire video is available, classification can be performed by processing the entire video. In the online case, where consecutive images are obtained, the objective is to make a decision using only a partial subset of the potentially available frames. For the sake of simplicity, we limit our discussion to the case of a binary classification.

**Standard offline case**    We are given a video $V = [V_0, \ldots, V_n]$, which is represented as a sequence of images $V_i$ (frames). Then, the network $\phi : \mathcal{V} \to [0, 1]$ returns the (soft) probability that $V$ belongs to the positive class. The final decision is based on the threshold $\tau$, where typically $\tau = 1/2$. Thus if $\phi(V) \geq \tau$, we conclude that the class of $V$ is positive.

An important observation is that given the knowledge what was the decision of the model for every threshold $\tau$, we can compute the probability $\phi(V)$.

**Proposition 3.1** *We have*

$$\phi(V) = \text{Prob}(\phi(V) \leq \tau : \tau \sim \text{unif}_{[0,1]}). \tag{1}$$

The above formula can be interpreted as selecting a random threshold value $\tau$ from the interval $[0, 1]$ and calculating the probability of being below the threshold. This is useful as in the case of online models, we have the natural definition of the threshold, and consequently it will allow to deduce the probabilistic model behind.

The subsequent paragraph addresses the question of how the offline model can be applied to the online procedure.

**Online (early exit) procedure**    We describe the standard general setting for the online early exit model. We assume that the frames arrive consecutively (on occasion, we permit them to arrive in groups of, e.g., two, four, or eight frames). Thereafter, given a trained classification network $\phi$ (as described in the previous paragraph), we fix a threshold $\tau \in [0, 1]$ and proceed with the following steps:

1. start with $k = 0$;

2. load $V_k$ and compute $p_k = \phi([V_0, \ldots, V_k])$;

3. if $p_k \geq \tau$, then return class 1 (positive) for $V = [V_0, \ldots, V_n]$; else put $k = k + 1$;

4. if $k = n + 1$, then stop the algorithm and return class 0 (negative); else return to step 2.

The objective of the aforementioned procedure is to allow making the decision of $V$ being in a positive class before loading all frames from $V$. It should be noted that if we operate in the offline mode, where we have access to all data, the above algorithm can be constrained to computing $p = \max\{p_1, \ldots, p_k\}$, and then determining that the class of $V$ is negative if $p < \tau$, or positive otherwise.

The underlying concept of our proposed solution is that of a continuous probabilistic model for the decision threshold of activity detection. This is discussed in detail in the following paragraph.

**Probabilistic model**    First, observe that once the aforementioned procedure is completed, we are only aware of the decision that has been made, but we lack the information necessary to calculate the probability of the given outcome. Without this information, it is not possible to use BCE loss and fine-tune the model. To address this issue, we apply the probabilistic concept from Proposition 3.1, which allows us to calculate the soft probability of the decision, given the knowledge that the decision was made for an arbitrary threshold. Then we get that the probability that $V$ has the positive class in the online case is given by

$$p(V) = \max(p_0, \ldots, p_n), \text{ where } p_i = \phi([V_0, \ldots, V_i]). \tag{2}$$

This is of paramount importance insofar as our objective is for the model to make more timely decisions (see Fig. 3).

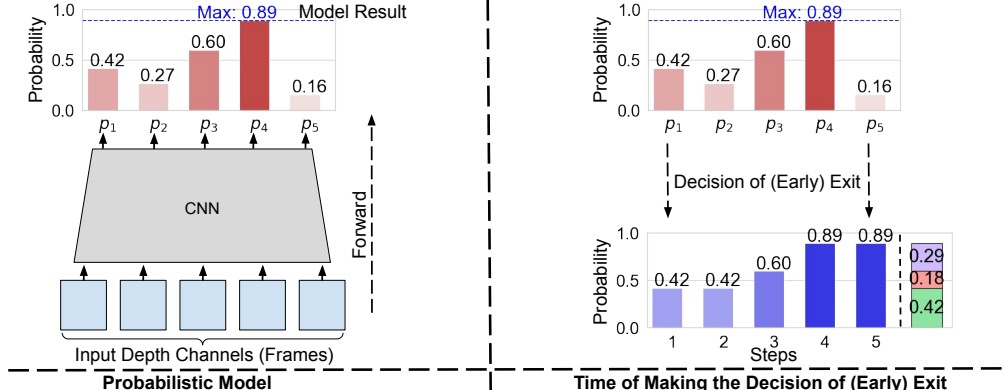

Figure 3: The left image shows a probabilistic model where the outcome is determined by selecting the maximum value of $p_i$ (in this case, 0.89). The right image shows an alternative approach where the model can exit earlier based on the available probabilities. In this example, the model terminates with a probability of 0.42 at the 1st step, a probability of 0.18 at the 3rd step, and a probability of 0.29 at the 4th step.

**Expected time of early exit**    To proceed to the probabilistic approach, we first calculate the expected time our model will make an early exit. For a fixed video $V$, let $T_V$ denote the random variable that returns the time we have made an early exit and $\infty$ when no early exit was made. We are interested in calculating the function $\text{Exit}(V)$, which represents the expected time of early exit (provided that it has been made). In order to achieve this, we define the random variable $W = T_V | T_V < \infty$, with values from the set $\{0, 1, \ldots, n\}$, as follows: we draw a random variable $\tau$ uniformly from the interval $[0, \max_i p_i]$, and for the given value of $\tau$, we return the first index $k \in [0, n]$ for which $p_k \geq \tau$. Then we have

$$\mathbb{E}(W) = p(W \geq 1) + \ldots + p(W \geq n) = n - P(V \leq 0) - \ldots - P(V \leq n - 1). \tag{3}$$

Finally, we get

$$\text{Exit}(V) = n - \frac{1}{\max p_i} \sum_{t=0}^{n-1} \max_{i=0..t} p_i \in [0, n] \quad \text{and} \quad \text{NE}(V) = \frac{1}{n}\text{Exit}(V) \in [0, 1]. \tag{4}$$

The function NE is defined as a normalization of $\text{Exit}(V)$ with respect to the number of frames. It returns 0 if we have made the exit (with probability one) on the frame $V_0$, and 1 if we cannot exit before $V_n$ and the probability of exit in $V_n$ is positive.

**Loss function for PrAViC**   Now we will describe how to incorporate the NE function into the loss function in a way that encourages the model to exit earlier. However, we do not want to stimulate the model too much so as not to provoke wrong decisions. To do this, we set a parameter $\lambda \in [0, 1]$ that tells us how large a percentage of confidence we are willing to potentially sacrifice in order to make the decision early. Consequently, we define the objective for image $V$ with class $y \in \{0, 1\}$ as follows:

$$\text{loss}_\lambda(V, y) = \text{BCE\_loss}(V, y) + y \log(\lambda + (1 - \lambda)\text{NE}(V)). \tag{5}$$

It is evident that for videos belonging to the negative class, the loss function remains the standard BCE loss.

For $\lambda = 1$, the additional part of the loss function is equal to 0, which discourages the model from making early decisions. As the value of $\alpha$ approaches 0, the model is encouraged to make decisions as rapidly as possible, even if this results in a loss of accuracy. We will mark the $\lambda$ for which the model was trained by using it as the subscript in PrAViC$_\lambda$.

**Remark 3.1** *Note that in Eq. (5) we do not penalize the points with negative class, in other words we do not encourage the model to make earlier decisions in this case. This follows from our motivation coming from real life situations, where class $1$ corresponds to an emergency-type event (heart or machine failure, car accident, etc.) and should be detected as early as possible, while class $0$ corresponds to the default (normal) state of the system.*

## 4   ARCHITECTURE OF THE MODEL

This section presents the modifications and extensions to the CNN-3D architecture that form the basis of our proposed approach. They include specific changes to key layers, including the convolution and batch normalization layers, as well as a unique method for processing the network's head. These modifications are designed to leverage the strengths of traditional CNNs while addressing specific challenges related to depth processing and feature extraction. The following paragraphs provide a detailed breakdown of each component and its role in our approach.

**Architecture for fine-tuning**   We outline the modifications made to the classic 3D CNN architecture to adapt it to our approach. While most components of a standard CNN architecture remain unchanged, we specifically alter the 3D convolution processing, batch normalization, and layer pooling. For 3D convolutions, we modify only those layers where the kernel size responsible for the depth (i.e., processing movie frames) is greater than 1. Our modification ensures that the kernels do not extend to the last deep channel. To achieve this, we replicate the input boundary on the front side before performing the multiplication operation, as illustrated in Fig. 4. For pooling layers, our modification involves replicating only the first depth channel.

The next essential mechanism in CNN networks is batch normalization (Ioffe & Szegedy, 2015). It involves calculating the mean and standard deviation for individual dimensions within mini-batches and training gamma and beta parameters during network training. This process ensures consistent and stable training across the network by maintaining the integrity of feature scaling and normalization, which is crucial for effective spatiotemporal pattern learning. In our approach, we modify this mechanism by statically determining the number of depth channels from which the statistics (mean and standard deviation) will be calculated. These statistics are then used to transform the remaining depth channels.

It is important to note that through these operations, each layer of our model only retains information from the preceding depth channels. In addition, our approach has a unique property: when provided with inputs of the same video, one containing $k$ frames and the other $n$ frames ($k < n$, where the

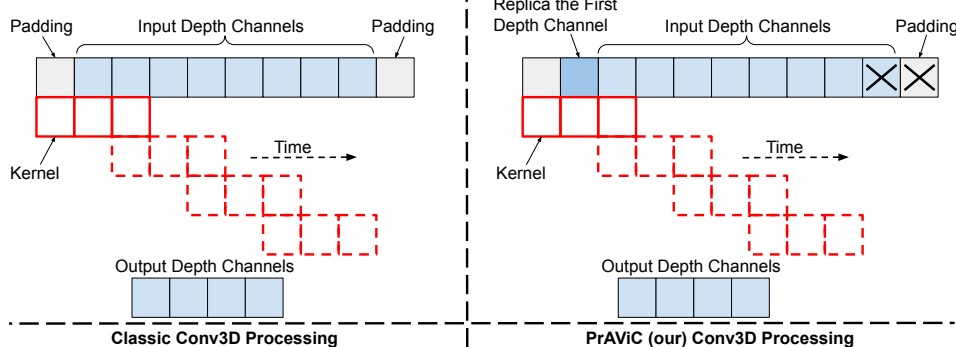

Figure 4: Illustration of the mechanism of the classical 3D convolution (left) with parameters: padding 1, stride 2, and kernel size 3, compared to our modified convolution (right). Note that the proposed change in input processing does not change the kernel weights.

second input is an extension of the first by $n - k$ frames), the network produces identical outputs, constrained to the dimensions of the output of the first $k$ depth channels of the inputs.

**Head design** Here we describe the modifications to the head of the CNN network that allow for online training and recursive evaluation, as shown in Fig. 5. Typically, in the case of a CNN, the head consists of the last linear layer, so in our approach we leave the head as a linear layer, but we will process the output of the last convolutional layer differently than in a classical CNN network. We assume that $v_0, \ldots, v_n \in \mathbb{R}^D$ represent the outputs of the last convolutional and pooling layers, considering only the height and width dimensions, while keeping the depth dimension as it is after the last convolutional layer. With this representation, following the standard offline approach, we perform a mean aggregation of the representations relative to time $t \in [0, \ldots, n]$:

$$w_t = \tfrac{1}{t+1} \sum_{i=0}^{t} v_i. \tag{6}$$

Using the aggregations $w_t$ above, we process each one separately through a linear layer $h$ followed by the sigmoid function $\sigma$ to obtain $p_t = \sigma(h(w_t))$. The final decision of the model is determined by the formula $p = \max_{t=0..n} p_t$. The source code is available at `https://github.com/...`

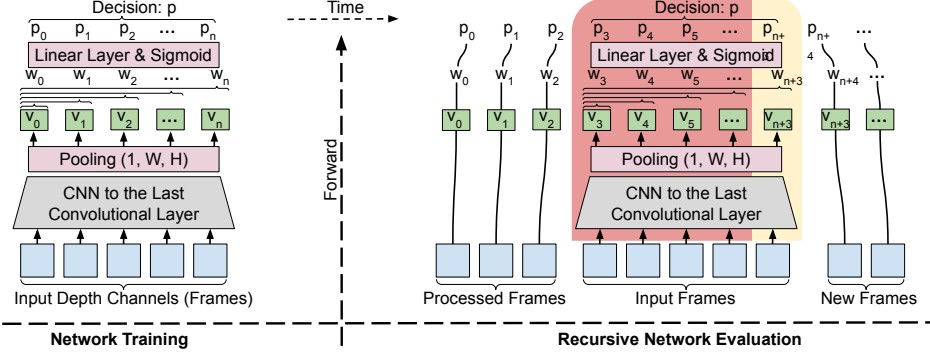

Figure 5: Illustration of our approach during the training phase (left) and the recursive evaluation phase (right). During evaluation, the red area retains computations from previous frames in individual network layers. In contrast, the yellow area represents computations for the recently introduced frame. Without retaining the computations in the red area, they would have to be recalculated, which would lengthen the evaluation process. This approach allows fast online processing of new frames.

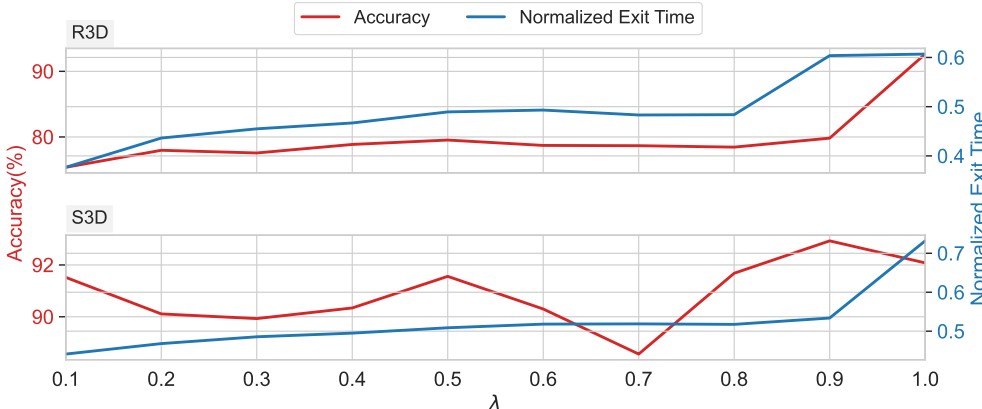

Figure 6: Comparison of accuracy and normalized exit time (NET) as a function of the custom loss function parameter $\lambda$. The top row shows results for PrAViC$_\lambda$ with the R3D-18 as a base model and the bottom row shows results for PrAViC$_\lambda$ with the S3D as a base model. Note that PrAViC tends to delay decisions and have higher accuracy as $\lambda$ parameter values increase, in contrast to faster decisions and lower accuracy observed for lower parameter values. Also, encouraging the model to make early decisions results in a small loss of accuracy. The base models, R3D and S3D, achieved accuracies of $94.41\%$ and $96.45\%$, respectively.

## 5 EXPERIMENTS

In this section, the experimental setup is detailed and the results that validate the effectiveness of the proposed approach are presented. Five datasets were employed for testing purposes, comprising four publicly available datasets (UCF101 (Soomro et al., 2012), EgoGesture (Zhang et al., 2018), Jester (Materzynska et al., 2019), and Kinetics-400 (Kay et al., 2017)), as well as one closed-access dataset (Ultrasound). The experiments employed a variety of architectural approaches, which are described in detail in the corresponding subsections. The numerical experiments were conducted using two different types of GPUs: NVIDIA RTX 4090 and NVIDIA A100 40GB.

**PrAViC vs. offline baselines**    In this paragraph, we undertake a comparative analysis of PrAViC with two non-online baseline approaches, namely ResNet-3D-18 (R3D-18) (Tran et al., 2018) and Separable-3D-CNN (S3D) (Xie et al., 2018). Experiments were conducted on the widely used video benchmark dataset UCF101 (Soomro et al., 2012).

From each video clip, 16 consecutive frames were extracted, starting with a random one. The R3D-18 model was resized to a resolution of $128 \times 171$, with each frame randomly cropped to $112 \times 112$. In contrast, the S3D model was resized to $128 \times 256$ resizing, with each frame randomly cropped to a size of $224 \times 224$.

Initially, the R3D-18 and S3D models underwent pre-training on UCF101. Subsequently, they were modified in accordance with the specifications outlined in Sec. 4, with the objective of transforming the offline models into online ones. In both cases, we employed SGD as the optimizer and cross-entropy as the loss function. The learning rates were set to 0.0002 for both of the offline models and to 0.002 and 0.0001 for the modified R3D-18 and S3D models, respectively. Lastly, a custom loss function, as defined in Eq. (5), was applied with the intention of forcing the model to make an earlier decision. The effectiveness of varying values for the $\lambda$ parameter, starting with 0.1 and increasing to 1, was evaluated.

Fig. 6 presents the accuracy and exit time (normalized to the interval [0,1]) obtained by the PrAViC$_\lambda$ modification (for the $\lambda$ parameter varying from 0.1 up to 1) applied to the considered baselines, namely R3D-18 and S3D. It is noteworthy that as $\lambda$ increases, both the accuracy and the exit time demonstrate a tendency to rise. Furthermore, our model ultimately attains a performance level that is nearly equivalent to that of the underlying offline models, while exhibiting a superior normalized exit time.

**PrAViC vs. online models** This paragraph presents a comparison between PrAViC and a collection of state-of-the-art online models, including 3D-SqueezeNet, 3D-ShuffleNetV1, 3D-ShuffleNetV2, and 3D-MobileNetV2 (Köpüklü et al., 2020). The models were trained on the EgoGesture (Zhang et al., 2018) and Jester (Materzynska et al., 2019) datasets, which have been specifically designed for the purpose of recognizing hand gestures from an egocentric perspective.

In setting up these experiments, the learned model weights (see (Köpüklü et al., 2020)) were utilized on the data. Initially, the model heads were trained anew, followed by light retraining of the entire model, with or without incorporation of an additional cost function to promote earlier detection. The architectures employed in our approach were identical to those utilized in the baseline models, with the requisite modifications detailed in Sec. 4. The models were trained using the SGD optimizer with standard categorical cross-entropy loss. The momentum, damping, and weight decay were set to 0.9, 0.9, and 0.001, respectively. The network learning rate was initialized at 0.1, 0.05, and 0.01, then decreased by a factor of 3 with a factor of 0.1 when the validation loss reached convergence.

During the training phase, input clips were selected from random time points within the video clip. If the video was composed of a smaller number of frames than the specified input size, a loop padding was incorporated. For the purpose of inputting data into the network, clips comprising multiples of 16 frames were utilized. In particular, 128 frames were used for 3D-MobileNetV2, while the remaining models utilized 192 frames. This approach was adopted to yield 8 probability values $p_i$, at the network output for the MobileNet network and 12 for the other networks. A single input clip possessed dimensions of $3 \times n \times 112 \times 112$, where $n$ represents the number of frames, which varied based on the models used (as described above).

Table 1: Comparison of top-1 (ACC@1) and top-5 (ACC@5) accuracy scores for PrAViC and baseline models trained on the EgoGesture and Jester datasets, demonstrating the improved performance of our approach and supporting online recursive exploitation.

| | Metric | SqueezeNet | | ShuffleNet | | ShuffleNetV2 | | MobileNetV2 | |
|---|---|---|---|---|---|---|---|---|---|
| | | Base | $PrAViC_1$ | Base | $PrAViC_1$ | Base | $PrAViC_1$ | Base | $PrAViC_1$ |
| EgoGesture | ACC@1 | 88.23 | **91.84** | 89.93 | **99.04** | 90.44 | **98.49** | 90.31 | **99.24** |
| | ACC@5 | 97.63 | **99.20** | 98.28 | **99.73** | 98.36 | **99.16** | 98.22 | **99.79** |
| Jester | ACC@1 | **90.74** | 90.52 | 93.08 | **94.48** | 93.69 | **95.45** | 94.34 | **95.25** |
| | ACC@5 | **96.75** | 95.99 | 99.50 | **99.53** | 99.57 | **99.59** | 99.63 | 99.62 |

Table 2: Top-1 (ACC@1) and top-5 (ACC@5) accuracy scores and normalized exit times (NETs) for PrAViC with and without the early decision cost function term (i.e., with the parameter $\lambda$ equal to 1 and 0.9), trained on the EgoGesture and Jester datasets. We observe that penalizing late decisions leads to reduced exit time without significantly affecting accuracy.

| | Metric | SqueezeNet | | ShuffleNet | | ShuffleNetV2 | | MobileNetV2 | |
|---|---|---|---|---|---|---|---|---|---|
| | | $PrAViC_1$ | $PrAViC_{0.9}$ | $PrAViC_1$ | $PrAViC_{0.9}$ | $PrAViC_1$ | $PrAViC_{0.9}$ | $PrAViC_1$ | $PrAViC_{0.9}$ |
| EgoGesture | NET | 0.5 | 0.3 | 0.7 | 0.4 | 0.7 | 0.4 | 0.6 | 0.1 |
| | ACC@1 | **91.84** | 90.86 | **99.04** | 98.70 | **98.49** | 97.00 | 99.24 | **99.58** |
| | ACC@5 | **99.20** | 99.16 | **99.73** | 99.66 | **99.16** | 99.08 | 99.79 | **99.92** |
| Jester | NET | 0.2 | 0.05 | 0.6 | 0.25 | 0.55 | 0.05 | 0.5 | 0.05 |
| | ACC@1 | **90.52** | 90.51 | **94.48** | 94.18 | **95.45** | 95.03 | 95.25 | **95.35** |
| | ACC@5 | **95.99** | 95.94 | **99.53** | 99.03 | **99.59** | 99.55 | 99.62 | **99.66** |

Tab. 1 provides a detailed comparison of the performance of PrAViC and the baseline models trained on the EgoGesture and Jester datasets, with the presentation of the top-1 and top-5 accuracy results

for the test set. As can be seen, PrAViC generally outperforms the other models across most metrics. These results demonstrate the efficacy of our approach and highlight its potential to deliver superior performance in online gesture recognition tasks. Additionally, Tab. 2 shows the results of our model with and without a cost function term for early decisions (see Eq. (5)). We can observe that penalizing late decisions leads to reduced exit time without significantly affecting accuracy.

**Ablation study** The objective of this paragraph is to investigate the impact of the proposed mean aggregation relative to time method on model performance and decision-making time. In particular, the influence of our method on the accuracy of the model and the exit time will be examined. The aim is to improve the model's ability to make timely predictions while maintaining or increasing accuracy by aggregating information across frames in a more structured way.

We conducted experiments using the CoX3D models introduced by Hedegaard & Iosifidis (2022), preserving the identical experimental setting. Our modification of the CoX3D network (variants S, M, and L) was evaluated on the test set of the Kinetics-400 dataset (Kay et al., 2017). One temporally centered clip was extracted from each video. The publicly accessible weights were utilized without any additional finetuning.

Table 3: The performance of the CoX3D and PrAViC X3D-based models trained on the Kinetics-400 dataset is evaluated. The accuracy and normalized exit time (NET) are calculated for the test set. It is observed that our solution, which aggregates the temporal information in a manner relative to time, exhibits superior performance.

| X3D Variant | Co Model | | $PrAViC_1$ Model | |
|---|---|---|---|---|
| | Accuracy (%) | NET | Accuracy (%) | NET |
| $X3D-S_{64}$ | 67.33 | 1 | **67.60** | 0.03 |
| $X3D-S_{13}$ | 60.18 | 1 | **66.43** | 0.27 |
| $X3D-M_{64}$ | 71.03 | 1 | **72.92** | 0.17 |
| $X3D-M_{16}$ | 62.80 | 1 | **70.42** | 0.59 |
| $X3D-L_{64}$ | 71.61 | 1 | **73.47** | 0.03 |
| $X3D-L_{16}$ | 63.03 | 1 | **69.05** | 0.27 |

The comparison results, comprising accuracy scores and normalized exit times (NETs), for the CoX3D and PrAViC X3D-based models are presented in Tab. 3. It should be noted that while the CoX3D models always make a decision on the last frame, PrAViC allows for the possibility of making a decision earlier with higher accuracy.

**PrAViC for medical use** In our final experiment, the details of which are presented in the appendix, we tested the PrAViC model on a real-life dataset of Doppler ultrasound images representing short-axis and suprasternal views of newborn hearts. The standard video analysis offline model, trained on this dataset, was modified by altering the classification layer (head) and the convolutional layers, with particular attention paid to the batch normalization layer. These modifications permitted the incorporation of subsequent frames in a sequential manner while the model was operational. Despite a slight reduction in accuracy (approximately 90%) in comparison to the offline model (94%), we were able to develop a model that reaches a final decision expeditiously (see Tab. 4 in the appendix).

## 6 CONCLUSIONS

In this work, we propose PrAViC, a general framework for automatically modifying networks that have been adapted for video processing to their online counterparts. The objective of PrAViC is to identify potentially dangerous situations at the earliest possible stage. To achieve this, we introduce a probabilistic theoretical model that underlies online data processing. We compute the mean expected exit time and use it as a component of the loss function to encourage the model to make early decisions. Furthermore, we propose a simple framework that translates offline models into online counterparts. It is demonstrated that we can encourage the model to make decisions earlier without a significant decrease in accuracy.

**Limitations**  In PrAViC, standard convolutions with a time component are employed. While this enables the utilization of pre-trained networks, it does not permit the comprehensive utilization of the full frames of the video in the residual network. This may potentially result in a minor loss of accuracy in comparison to the offline model.

**Broader impact**  As the utilization of online video analysis becomes increasingly prevalent in both medical and social contexts, our methodology can be employed as a straightforward instrument for researchers engaged in applied video processing.

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

## A    APPENDIX

This section presents additional results that did not fit into the main paper. These supplementary findings provide further insights and reinforce the conclusions drawn from our primary analyses. The results discussed here include analytical approaches and complementary experiments that offer a broader perspective on the research topic.

### A.1    EXPERIMENT ON THE ULTRASOUND DATASET

We conducted an experiment, in which the PrAViC model was tested on a real medical dataset of Doppler ultrasound images representing short-axis and suprasternal views of newborns' hearts. These recordings were obtained as part of an ongoing scientific research project involving pediatric cardiologists, with the consent of the newborns' parents. During the acquisition process, a total of 18,365 ultrasound recordings were collected. The standard video analysis offline model trained on this dataset was modified by changing the classification layer (head) as well as the convolutional layers, in particular batch norm. These changes, which required fine-tuning of the model, allowed for subsequent frames to be added sequentially while the model was running. Since the classification layer was completely new, fine-tuning started with 5 epochs of learning only this layer with a learning rate of $10^{-5}$. Subsequent training of this model for 10 epochs, with the same learning rate, but this time with training of all changed layers, allowed to achieve accuracy of $94\%$. Standard Cross-entropy as a cost function was used in both stages.

Using our own cost function provided in Eq. (5) (see the main paper), which prefers to classify as a class 1 element as quickly as possible, if it belongs to it, allowed us to achieve slightly lower accuracy as in the standard model, so approximately $90\%$ regardless of the lambda parameter (see Tab. 4). Due to the use of remembered history, the model accuracy was tested by introducing 16 frames into the model, i.e., the number needed for each convolutional layer inside the model to have at least one output representation other than padding. Only subsequent frames were added one at a time.

The model was then repeatedly evaluated on the selected video with different frame delivery parameters. The average time from 100 evaluations was taken as the result of the experiment for each set of parameters. The first parameter was the total number of frames delivered after all steps were completed. Due to the growing history in memory, it was expected that the increase in time would not be linear, i.e., 2 times as many frames should result in an evaluation more than 2 times longer. The experimental results confirmed this thesis. On the other hand, not single frames but entire batches can be loaded into the model. For this reason, in the second experiment, the first experiment was repeated, but instead of a single frame, 4 or 8 frames were fed to the model in each step. Thus, the history stored in the model

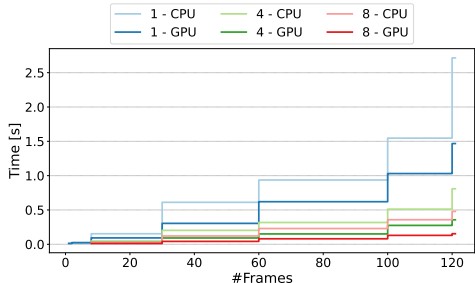

Figure 7: Model evaluation on time. Each line corresponds with the device used for evaluation and the number of frames inserted into the model in one step.

worked in the same way as before, but the number of reads and writes to it was reduced. Due to the expected operation of the model on devices with lower performance, e.g., mobile devices, the first part of the tests was carried out on a CPU. The experiments were also repeated using the CUDA architecture. The results from both devices are presented in Fig. 7.

The conducted experiments have demonstrated the efficacy of the developed methodology in identifying congenital heart defects (CHDs) in neonates through ultrasound imaging. CHDs, encompassing conditions such as Tetralogy of Fallot, Hypoplastic Left Heart Syndrome, and Ventricular Septal Defect, pose formidable diagnostic hurdles owing to their intricate nature and the subtleties inherent in early cardiac anomalies. Regrettably, undetected instances of such defects represent a prominent contributor to neonatal mortality rates. The adapted video analysis model utilized in this investigation heralds a significant leap forward in the diagnostic realm pertaining to congenital heart defects.

Table 4: Classification accuracy results obtained by PrAViC$_\lambda$ on the Ultrasound dataset. Note that the model does not show much difference in accuracy depending on the $\lambda$ parameter, since most decisions are made after processing the first 16 frames anyway.

| $\lambda$ | 0.1 | 0.2 | 0.3 | 0.4 | 0.5 | 0.6 | 0.7 | 0.8 | 0.9 | 1.0 |
|---|---|---|---|---|---|---|---|---|---|---|
| **Accuracy(%)** | 86.95 | 88.53 | 88.53 | 90.90 | 90.90 | 91.30 | 91.30 | 90.90 | 90.90 | 88.93 |

## A.2 STUDY OF NE FUNCTION DURING TRAINING

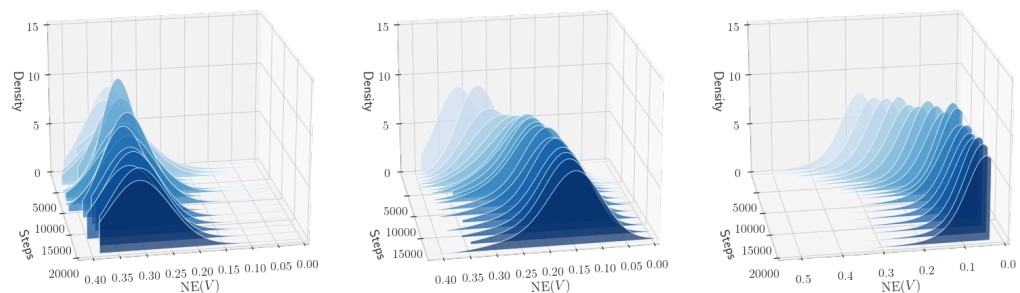

Figure 8: Illustration of the progression of changes in the NE function during 3D-SqueezeNet training process (from left to right) using the cost function given in Eq. (5) (see the main paper) for $\lambda = 1, 0.9$, and $0.5$, respectively. For $\lambda = 1$, where late detection of classes is not penalized, the NE value remains relatively stable. However, as the $\lambda$ parameter decreases, the NE value shows a more noticeable decline, indicating the increasing impact of penalizing late detection on the model performance.

Based on the 3D-SqueezeNet model, we show the course of changes in the NE function (see Eq. (4) in the main paper) while training this model. The image demonstrates these changes using the cost function given in Eq. (5) for $\lambda = 1, 0.9$, and $0.5$, respectively. As depicted in Fig. 8, with $\lambda = 1$, the NE value remains relatively stable, whereas a reduction in the $\lambda$ parameter results in a more noticeable decline in the NE value, highlighting the effect of penalizing late detection on model performance.

Fig. 9 presents histograms of decisive frame numbers for PrAViC$_\lambda$ with the R3D-18 (top row) and S3D (bottom row) baseline models (see Sec. 5 in main paper), comprising different values of the $\lambda$ parameter. In both cases, we observe that larger $\lambda$ values (depicted in deep blue shades) result in higher decisive frame numbers. Conversely, smaller $\lambda$ values (depicted in white and light blue shades) lead the model to make decisions much earlier, as expected.

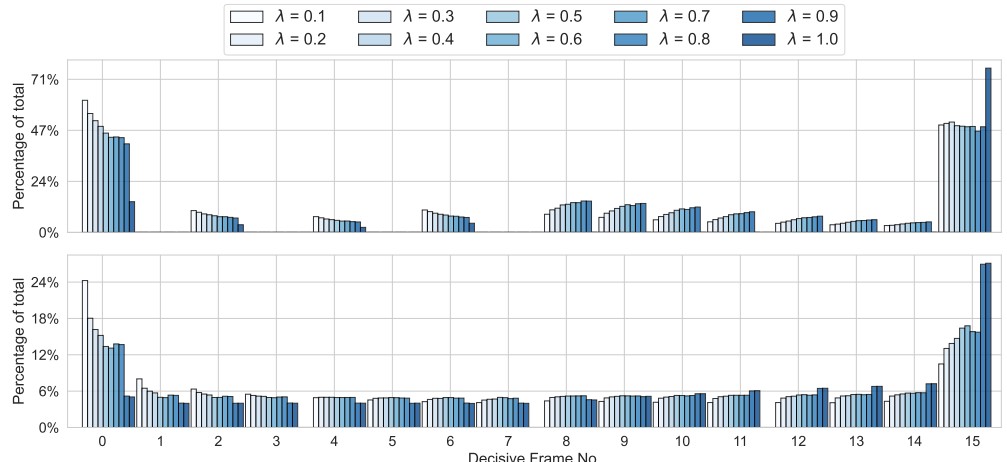

Figure 9: Comparison of histograms of decisive frame numbers for the R3D-18-based (top row) and S3D-based (bottom row) PrAViC models with different loss function $\lambda$ parameters, varying from 0.1 to 0.9. The higher the $\lambda$ value, the higher the percentage of higher numbers of decisive frames. Decreasing the $\lambda$ parameter encourages the model to make decisions earlier.

