# OpenReview forum: "PrAViC: Probabilistic Adaptation Framework for Real-Time Video Classification"
_ICLR.cc/2025/Conference — Submitted to ICLR 2025_

### Official Review · Reviewer_FeKn · 2024-10-16

**Soundness:** 2
**Presentation:** 3
**Contribution:** 2
**Rating:** 3
**Confidence:** 4

**Summary:**

The authors propose PrAViC, a framework for early (online) video classification. PrAViC incorporates a stopping mechanism that allows the model to exit early and make decisions based on partial video frames, thereby improving classification speed while maintaining high accuracy. The framework tracks the maximum class probability during frame processing and exits once a certain confidence threshold is reached, predicting the class with the highest probability at the time of exit.

Please note that the following represents my initial impressions of the paper, and I am open to discussion. I welcome any corrections to potential misunderstandings.

**Strengths:**

- **Early classification:** PrAViC makes faster decisions by exiting early when confident, a crucial feature for applications like emergency detection, medical diagnostics, and autonomous systems.
- **Minimal accuracy loss:** The framework maintains accuracy comparable to offline models, even with early exits.
- **Adaptability:** PrAViC can be integrated with existing 3D CNN architectures and easily adapted for real-time operation with minimal modifications.
- **Extensive experiments:** The paper presents experiments across five real-world datasets.
- **Clear visual aids:** Figures such as Fig. 3, 4, and 5 effectively illustrate the proposed model, aiding comprehension.

**Weaknesses:**

The paper was an enjoyable read, but I found the following potential weaknesses and questions. I am willing to raise my score once these issues are addressed.


**Lack of highly related baselines.**
The work is positioned as a variant of Early Classification of Time Series (ECTS), especially given the focus on “real-time processing” and early decision-making. However, there is no mention or evaluation of ECTS algorithms [1, 2, 3], making it difficult to understand how PrAViC compares to existing work. ECTS literature often employs loss functions that balance classification accuracy and timeliness, much like the approach taken in Eq. (5).

Several existing early classification models, such as video early classification [4], could be applied to the problem setting defined by the authors. Although many ECTS algorithms are feature-vector-based [5, 6, 7] and not intended for video stream per se, they can also be applied to the aggregated feature in Eq.(6) under PrAViC framework.

To justify the novelty and effectiveness of their loss function, the authors should compare PrAViC against these established methods.  While the proposed framework demonstrates solid results across diverse real-world scenarios, evaluating against related baselines would strengthen its position.

[1] Xing+, Early prediction on time series: A nearest neighbor approach. IJCAI 2009
[2] Gupta+, Approaches and applications of early classification of time series: A review. IEEE TAI 2020.
[3] Mori+, Reliable early classification of time series based on discriminating the classes over time. DMKD 2016
[4] Ma+, Learning activity progression in lstms for activity detection and early detection. CVPR 2016
[5] Chen+, Early time series classification using TCN-transformer. ICCASIT 2022
[6] Hartvigsen+, Adaptive-halting policy network for early classification. KDD 2019
[7] Bilski+, CALIMERA: A new early time series classification method. Information Processing & Management 2023


**Lack of proper evaluation for online classification.**

While early exit or ECTS models should be evaluated on *both* classification performance and earliness (such as NET), some experiments report only accuracy (e.g., Table 1), or only PrAViC's NET (e.g., Table 2). Ideally, early exit models should not be evaluated based on a single point in the accuracy-NET space; rather, they should be assessed using a set of points that represent the tradeoff between accuracy and earliness (i.e., the Pareto front). Benchmarking the proposed model and the baselines across this tradeoff would provide a more comprehensive evaluation, showcasing how well the model balances early exits and accuracy, which is crucial for early classification tasks.

Additionally, given the real-world datasets used, which often suffer from class imbalance, accuracy may be misleading [8]. More appropriate metrics would be balanced accuracy or mean macro-averaged recall, which would offer a clearer picture of the model’s performance across all classes.

[8] Branco+, A survey of predictive modeling on imbalanced domains. ACM Comput. Surv 2016


**Potentially overstated theoretical contribution.**

The authors claim that PrAViC is a theoretically-based framework, contrasting it with "hand-devised methods." However, I found no formal theoretical guarantees such as convergence rates, statistical consistency, or classification-calibrated loss. If the "theoretically-based" claim stems solely from the thresholding mechanism using class probabilities, this risks misleading readers and undermining the significance of PrAViC’s novelty.

Furthermore, there’s no engagement with well-established literature on Sequential Hypothesis Testing [9, 10, 11], which already provides theoretical guarantees for similar problems. Including a comparison or discussion of this body of work would strengthen the theoretical claims of the paper.

Thus, to substantiate the claim of being "theoretically-based," the authors could strengthen their work by providing formal guarantees, similar to what is done in Sequential Probability Ratio Test (SPRT) [9]. For example, SPRT guarantees the minimal number of samples (or equivalently, exit time) under a predefined error rate, assuming certain statistical conditions. If PrAViC could offer a theoretical guarantee of exit time (or the balance between exit time and error rate) under similar assumptions, it would provide a strong theoretical foundation and significantly enhance the contribution of the paper. Such guarantees would better differentiate PrAViC from "hand-devised" methods and provide formal support for its early exit mechanism.

[9] Wald, Sequential tests of statistical hypotheses. Annals of Mathematical Statistics, 1945
[10] Tartakovsky. Asymptotic optimality of certain multihypothesis sequential tests: Non-i.i.d. case. Statistical Inference for Stochastic Process, 1998
[11] Miyagawa+, The power of log-sum-exp: Sequential density ratio matrix estimation for speed-accuracy optimization. ICML, 2021


**Inappropriate ablation study**

The ablation study is not conducted in the conventional sense. Typically, I would expect the ablation study to evaluate the contribution of individual components (e.g., removing batch normalization, disabling the loss term). However, the authors introduce a new model, CoX3D, and conduct new experiments, which is not in line with what is usually meant by an ablation study. This adds confusion and detracts from the clarity of the experimental analysis.


**Reproducibility concerns.**

The code for PrAViC is mentioned but not provided (there’s a broken link to GitHub), raising concerns about reproducibility. Furthermore, the authors report only scalar accuracy values, without error bars or statistical tests, making it difficult to evaluate the robustness and generalizability of their results. For a complex training process, such as "light retraining" of models, more details should be provided to ensure that others can replicate the experiments.


**Unclear PrAViC framework for multiclass classification.**

Section 3 formulates the problem in a binary classification setting, but it’s not clear how the framework extends to K-class classification (e.g., UCF101 with 101 classes). Are the authors treating it as a K+1-class problem (one background class and K event classes), or are they performing standard K-class classification without a background class? Clarifying this would enhance the understanding of the framework's flexibility in real-world, multiclass scenarios.


**Unclear motivation and novelty.**
The authors state that PrAViC is motivated by the need for "more generalizable solutions that can accommodate diverse forms of data." However, the terms "generalizable" and "diverse forms of data" are not clearly defined in the paper. It would be beneficial to clarify whether "generalizable" refers to performance across different datasets or some specific capability of PrAViC that makes it more adaptable than existing online classification methods.

Additionally, "diverse forms of data" seems to refer to the range of real-world datasets (video streams) used in the experiments. However, since the paper only addresses video classification, it would help to rephrase or elaborate on this claim to avoid confusion. Classical CNN-RNN architectures [12] already perform online video classification effectively, so it would also be useful to highlight the specific contributions PrAViC offers that go beyond these established methods.

[12] Karpathy+, Large-scale Video Classification with Convolutional Neural Networks, CVPR 2014


**Minor comments**

- l.063 “two publicly available datasets” → should be "four"?
- When I saw Fig.2 for the first time, I had no idea what does NE(V) stands for. I would add a definition or at least reference in the legend, but this may be a personal preference.
- The term "unified" is used repeatedly, but it’s unclear what aspect of PrAViC is unified compared to the literature.
- l.178 “BCE” loss is introduced without spelling out. I can tell it’s a binary cross entropy-loss, but please avoid letting a reader guess.
- l.203~ “In this example, the model terminates with a probability of 0.42 at the 1st step, a probability of 0.18 at the 3rd step, and a probability of 0.29 at the 4th step.” → should be 0.60 at the 3rd, and 0.89 at the 4th?
- Figure 6: The caption should specify which dataset is used (UCF101).
- Figure 6: Adding a horizontal line to show the baseline model performance would help clarify the comparison.

**Questions:**

I incorporated my questions to the weaknesses section. Please see the above.

**Details Of Ethics Concerns:**

N.A.

---

> ### Comment · Reviewer_FeKn · 2024-11-22
>
> With the discussion period nearing its conclusion, I wanted to kindly check if the authors plan to address my comments. As I mentioned, my feedback reflects an initial impression, and I remain open to further discussion. If you intend to engage, I encourage doing soon, as the final days are likely to be busy with my own rebuttal process.

---

### Official Review · Reviewer_ZLKn · 2024-11-01

**Soundness:** 3
**Presentation:** 2
**Contribution:** 3
**Rating:** 5
**Confidence:** 3

**Summary:**

This paper proposes a probabilistic adaptation framework for real-time video classification. The framework is a probabilistic approach that encourages the model to decide whether the instance belongs to the positive class as soon as possible. Besides, the authors also propose a series of adaptations to transform an offline video classification model into online classification. The authors evaluate the framework on four open datasets: UCF101, EgoGesture, Jester, and Kinetics-400, considering different 3D-CNN architectures. The experiments show that the proposed framework (and architectural modifications) can adapt offline models to online models and that the models can make decisions faster than offline models.

**Strengths:**

- Good and helpful images.
- The probabilistic adaptation framework is a clever idea to handle the need for classification as fast as possible.
- The architectural modifications are simple and easy to implement in any offline CNN structure.
- The experiments cover a sufficient range of datasets and relevant 3D-CNN models to show the effectiveness of the proposed framework.
- Comparison with online and offline models.
- Another method for online video classification is used for comparison.

**Weaknesses:**

- The paper does not mention the SOTA models for video classification and action recognition, such as VideoMAE and other transformer-based architectures. Transformer models are a well-established SOTA, especially for some of the evaluated datasets.
- The paper discusses only 3D-CNNs and does not clarify such a limitation in the abstract or contributions. The framework's adaptations do not include different architectures, such as transformers or even CNN-LSTM.

Papers:
VideoMAE V2: Scaling Video Masked Autoencoders with Dual Masking
Bidirectional Cross-Modal Knowledge Exploration for Video Recognition with Pre-trained Vision-Language Models
OmniVec2 - A Novel Transformer based Network for Large Scale Multimodal and Multitask Learning

- The explanation of the Expected Time of Early Exit is not very clear. The term P(V = x) is used before explaining P in the text, making it unclear for the reader if they are referring to Video V or some property of it. (Eq. 3, Pg. 4)

- Authors do not provide a GitHub repo; only provide an indication that it will be released. It could be anonymized and provided.

**Questions:**

- According to the paper's premise of using the model for real-world situations, two analyses would have been interesting: analyzing the model's false positive rate and how it changes according to the lambda hyperparameter and the impact of the early exit on the overall prediction time (in seconds).
- A term alpha appears in the middle of the text and it is never shown. I suppose that is a mistake and should be the hyperparameter lambda. (Above Eq. 5, Pg.5)
- In the abstract, the authors mention that they will evaluate three datasets with two of them publicly available. They mention four open datasets, as they also correctly explain in the experiments.

---

### Official Review · Reviewer_9vGq · 2024-11-02

**Soundness:** 2
**Presentation:** 2
**Contribution:** 2
**Rating:** 3
**Confidence:** 4

**Summary:**

This paper introduces a novel scheme for online video classification. Such a method is based on the training/finetuning of offline models with a regularization term that should drive the classification network to give early classification, defining a tradeoff between computation and accuracy. The method is tested by equipping previously presented architecture with such new loss.

**Strengths:**

- The related work section is well presented and easy to read
 - Section 3, up to “Expected time of early exit”, gives a clear overview of the problem that the paper wants to address

**Weaknesses:**

- the paper is generally not clear from section 3 on
 - the proposed approach is never well defined. a custom loss function is proposed, but it is never said how to then use the model to do inference with early exit. Instead the paper would benefit from a comparison between a baseline model, with a fixed threshold, and the proposed finetuned ones, to compare the time the two models take to classify the video as 0/1 above the significance threshold picked
 - the theoretical part is highly confusing, with missing explanations and equations and quantities that are never used. in particular, all of these concerns are reported in the “Problems” section of this review.
 - the paper misses a section comparing the proposed approach to any baseline for online classification, even basic ones reported in “Online (early exit) procedure”
 - The paper is very practical, with little and very confusing theoretical contributions. For this reason, it is fundamental that the paper is presented with a reproducible code, which would also help the understanding of the not-so-clear “Expected time of early exit”. Particularly, ICRL strongly encourage to add a “Reproducibility Statement” (https://iclr.cc/Conferences/2025/AuthorGuide), and the paper is missing it.

**Questions:**

- Eq.3 is defined and never used. Furthermore, it’s not clear how the expected value is computed as reported. Generally speaking, you define W, and never use it anywhere.
 - Eq.3 clearly reports $P(V\le c)$, which makes no sense, since $V$ is defined as a set of frames
 - It is not clear, and as previously stated, not defined, how Eq.3 and Eq.4 are defined. But more generally speaking, the whole “Expected time of early exit” section should be rewritten as definitely not clear, which is a problem since it lays the ground for the rest of the paper. Indeed, by the looks of things, what you are doing with Exit(V) is just the integral of the blue  histogram in Fig 3, though this is a way more intuitive way to explain the term, over the notation used in the section
 - in Eq.5 it is not clear why you decided to use a log scale for the regularization. More generally, the equation would benefit from a more extensive explanation, showing the intuition behind it
 - in line 375, it is said that accuracy increases, though in the S3D case, that’s opinable observing the plot. Ideally, if it’s still the case that the authors want to state that, please run a statistical analysis to check if any relevant trend can be seen in such a plot and report the p values somewhere in the paper.
 - in line 378, you state that you _compare_ your approach to many others, though, from Table 1, it seems more like that you add your approach on top of such methods. At no point, however, you state what you mean when you refer to “baseline” in Table 1
 - in table 1, ignoring that the authors never define what “baselines” means, there are some results that increase the accuracy by 10%, which is almost-99% error rate. however the results are compared to the setting lambda=1 which means that the loss is composed of only the BCE loss. However, that term is the main contribution of the paper. Thus, it’s not clear how it is possible to have such a bump in performance.
Writing concerns:
 - line 152 “This is useful as in the case of online...” might need a rephrasing
 - line 162, since you are using “=” for comparison and assignment, i’d suggest to use “:=” for assignment, just for clarity
 - line 234, you are using alpha instead of lambda
 - line 304 contains an empty link
 - Figure 5 contains a random “3" behind the word “sigmoid” on the right
 - line 383, there is a citep that should be a normal cite
 - the paper contains only very long main sections (\section). Instead, it’s highly suggested the usage of \subsections, \subsubsection, and so on to structure the paper in a more reasonable way

---

### Official Review · Reviewer_ym1U · 2024-11-04

**Soundness:** 3
**Presentation:** 3
**Contribution:** 3
**Rating:** 6
**Confidence:** 3

**Summary:**

The paper proposed a method of an online video classification by using the early-exit mechanism. The solution is reasonable and the results is impressive.

**Strengths:**

1. The propose solution is reasonable and the paper is well written
2. The experiment results are impressive, which may help for other sub-sequent tasks

**Weaknesses:**

1. more baselines should be added to validate the effectiveness of the proposed approach. For example, some other on-line mechanisms, even the simple threshold one.
2. Some discussion should be added to explain the intuition of the method, and more ablation study should be added to validate the findings, and the contributions.

**Questions:**

See weakness

---

### Meta-Review · Area_Chair_L5xe · 2024-12-21

**Metareview:**

This paper was reviewed by four experts in the field and received 6, 3, 5, 3 as the ratings. The reviewers agreed that the proposed PrAViC method enables faster video classification by exiting early when confident, which can be useful in a variety of applications; it can be easily integrated with existing 3D CNN architectures, and has been validated on 5 real-world datasets.

The reviewers raised concerns about the comparison baselines, stating that PrAViC has not been compared against related algorithms for Early Classification of Time Series (ECTS); it is difficult to assess the merit of the framework without an exhaustive comparison against the state-of-the-art methods. While the authors claimed that PrAViC is a theoretical framework, no formal theoretical guarantees (such as convergence rates, statistical consistency, or classification-calibrated loss) were provided in the paper. Concerns were also raised about the ablation studies, where new experiments were conducted with a new model, rather than the conventional method of evaluating the contribution of each individual component. The writing quality of the paper also needs to improve. The authors are also encouraged to open-source their implementation for better reproducibility (the provided GitHub link is broken).

The authors did not respond to the individual reviewer's comments. In light of the above discussions, we conclude that the paper is not ready for an ICLR  publication in its current form. While the paper clearly has merit, the decision is not to recommend acceptance. The authors are encouraged to consider the reviewers' comments when revising the paper for submission elsewhere.

**Additional Comments On Reviewer Discussion:**

Please see my comments above.

---

### Decision · Program_Chairs · 2025-01-22

Reject